

# Evaluating graph neural networks under graph sampling scenarios

Qiang Wei[1,2] and Guangmin Hu[1]

[1] University of Electronic Science and Technology of China, School of Information and Communication Engineering, Chengdu, Sichuan, China

[2] National Key Laboratory of Science and Technology on Blind Signal Processing, Chengdu, Sichuan, China

## ABSTRACT

**Background**. It is often the case that only a portion of the underlying network structure is observed in real-world settings. However, as most network analysis methods are built on a complete network structure, the natural questions to ask are: (a) how well these methods perform with incomplete network structure, (b) which structural observation and network analysis method to choose for a specific task, and (c) is it beneficial to complete the missing structure.

**Methods**. In this paper, we consider the incomplete network structure as one random sampling instance from a complete graph, and we choose graph neural networks (GNNs), which have achieved promising results on various graph learning tasks, as the representative of network analysis methods. To identify the robustness of GNNs under graph sampling scenarios, we systemically evaluated six state-of-the-art GNNs under four commonly used graph sampling methods.

**Results**. We show that GNNs can still be applied on single static networks under graph sampling scenarios, and simpler GNN models are able to outperform more sophisticated ones in a fairly experimental procedure. More importantly, we find that completing the sampled subgraph does improve the performance of downstream tasks in most cases; however, completion is not always effective and needs to be evaluated for a specific dataset. Our code is available at https://github.com/weiqianglg/evaluate-GNNs-under-graph-sampling.

Corresponding author
Qiang Wei,
weiqiang@std.uestc.edu.cn,
weiqianglg@163.com

# INTRODUCTION

In the last few years, graph neural networks (GNNs) have become standard tools for learning tasks on graphs. By iteratively aggregating information from neighborhoods, GNNs embed each node from its $k$-hop neighborhood and provides a significant improvement over traditional methods in node classification and link prediction tasks (*Dwivedi et al., 2020*; *Shchur et al., 2018*). Powerful representation capabilities have led to GNNs being applied in areas such as social networks, computer vision, chemistry, and biology (*Hou et al., 2020*). However, most GNN models need a complete underlying network structure, which is often unavailable in real-world settings (*Wei & Hu, 2021*).

Frequently it is the case that only a portion of the underlying network structure is observed, which can be considered as the result of graph sampling (*Al Hasan, 2016*; *Ahmed, Neville & Kompella, 2013*; *Blagus, Šubelj & Bajec, 2015*; *Dwivedi et al., 2020*; *Hu & Lau, 2013*). Graph sampling has become a standard procedure when dealing with massive and time evolving networks (*Ahmed, Neville & Kompella, 2013*). For example, on social networks such as Twitter and Facebook, it is impossible for third-party aggregators to collect complete network data under the restrictions for crawlers, we can only sample them by various different users. Unfortunately, many factors make it difficult to perform multiple graph sampling. First, the time consuming, communication networks such as the Internet need hours or days to be probed (*Ouédraogo & Magnien, 2011*). Moreover, measuring the network structure is costly, *e.g.*, experiments in biological or chemical networks. Graph sampling scenarios bring an additional challenge for GNNs, and little attention has been paid to the performance of GNN models under graph sampling.

In this experimental and analysis paper, we consider the observed incomplete network structure $G_O$ as one random sampling instance from a complete graph $G$, then we address the fundamental problem of GNN performance under graph sampling, in order to lay a solid foundation for future research. Specifically, we investigate the following three questions:

**Q1:** Can we use GNNs if only a portion of the network structure is observed?

**Q2:** Which graph sampling methods and GNN models should we choose?

**Q3:** Can the performance of GNNs be improved if we complete the partial observed network structure?

To answer the above questions, we design a fairly evaluation framework for benchmarking GNNs under graph sampling scenarios by following the principles in *Dwivedi et al. (2020)*. Specifically, we performed a comprehensive evaluation of six prominent GNN models under four different graph sampling methods on eight different datasets with three semi-supervised network learning tasks, *i.e.*, node classification, link prediction and graph classification. The GNN models we implemented include Graph Convolutional Networks (GCN) (*Kipf & Welling, 2017*), GraphSage (*Hamilton, Ying & Leskovec, 2017*), MoNet (*Monti et al., 2017*), Graph Attention Network (GAT) (*Veličković et al., 2017*), GatedGCN (*Bresson & Laurent, 2017*), and Graph Isomorphism Network (GIN) (*Xu et al., 2018*), and the graph sampling methods we used include breadth-first search (BFS), forest fire sampling (FFS), random walk (RW) and Metropolis–Hastings random walk (MHRW). Our main findings are summarized as follows:

- In most single graph datasets, we can still use GNNs under graph sampling scenarios if the sampling ratio is relatively large; however, sampling on multi-graph datasets causes GNNs to fail.
- The best GNN model and sampling method are GCN and BFS in small datasets, GAT and RW in medium datasets, respectively.
- In most cases, completing a sampled subgraph is beneficial to improve the performance of GNNs; but completion is not always effective and needs to be evaluated for a specific dataset.

As far as we know, this is the first work to systematically evaluate the impact of graph sampling on GNNs.

## RELATED WORK

In this section, we briefly review related works on graph sampling and GNNs.

### Graph sampling

Graph sampling is a technique to pick a subset of nodes and/or edges from an original graph. The commonly studied sampling methods are node sampling, edge sampling, and traversal-based sampling (*Al Hasan, 2016*; *Ahmed, Neville & Kompella, 2013*). In node sampling, nodes are first selected uniformly or according to some centrality, such as degree or PageRank, then the induced subgraph among the selected nodes is extracted. In edge sampling, edges are selected directly or guided by nodes. Node sampling and edge sampling are simple and suitable for theoretical analysis, but in many real scenarios we cannot perform them due to various constraints, *e.g.*, the whole graph is unknown (*Hu & Lau, 2013*). Traversal-based sampling, which extends from seed nodes to their neighborhood, is more practical. Therefore, a group of methods was developed, including breadth-first search (BFS), depth-first search (DFS), snowball sampling (SBS) (*Goodman, 1961*), forest fire sampling (FFS) (*Leskovec, Kleinberg & Faloutsos, 2005*), random walk (RW), and Metropolis–Hastings random walk (MHRW). With the numerous graph sampling methods developed, the question of how they impact GNNs still remains to be answered.

### GNNs

After the first GNN model was developed (*Bruna et al., 2014*), various GNNs have been exploited in the graph domain. GCN simplifies ChebNet (*Defferrard, Bresson & Vandergheynst, 2016*) and speeds up graph convolution computation. GAT and MoNet extend GCN by leveraging an explicit attention mechanism (*Lee et al., 2019*). Due to powerful represent capabilities, GNNs have been applied into a wide range of applications including knowledge graphs (*Zhang, Cui & Zhu, 2020*), molecular graph generation (*De Cao & Kipf, 2018*), graph metric learning and image recognition (*Kajla et al., 2021*; *Riba et al., 2021*). Recently, graph sampling was investigated in GNNs for scaling to larger graphs and better generalization. Layer sampling techniques have been proposed for efficient mini-batch training. GraphSage performs uniform node sampling on the previous layer neighbors (*Zeng et al., 2019*). GIN extends GraphSage with arbitrary aggregation functions on multiple sets, which is theoretically as powerful as the Weisfeiler–Lehman test of graph isomorphism (*Xu et al., 2018*). In contrast to layer sampling, GraphSAINT constructs mini-batches by directly sampling the training graph, which decouples the sampling from propagation (*Zeng et al., 2019*). However, in most GNNs it is assumed that the underlying network structure is complete without data loss, which is often not the case.

In addition, different GNNs are compared in *Errica et al. (2019)* and *Shchur et al. (2018)* with regard to node classification and graph classification tasks, respectively, a systematic evaluation of deep GNNs is presented in *Zhang et al. (2021)*, and a reproducible framework

for benchmarking of GNNs is introduced in *Dwivedi et al. (2020)*. The most related work to ours is *Fox & Rajamanickam (2019)*, in which the robustness of GIN to additional structural noise is studied. Our work focuses on graph sampling that can be considered as a random structure removed from the original network.

## Models

We focus on the robustness of GNNs under graph sampling scenarios. As shown in Fig. 1, $G_O$ is the partial observed graph from a network $G$, which is often difficult to make complete observations. We train GNNs on $G_O$ and then evaluate on three typical learning tasks: node classification, link prediction and graph classification. In this paper, we treat $G_O$ as one of the many graphs generated by a certain sampling process from a known $G$, consequently we are able to determine the robustness of GNNs in a statistical way via multiple independent random sampled $G_O$.

We denote the original network as $G(V, E, X)$, where $V$ and $E$ represent node and edge sets, respectively, and $X \in \mathbb{R}^{|V| \times d}$ denotes the attribute matrix. There is no missing structure in $G$. The observed or sampled graph is represented by $G_O(V_O, E_O, X_O)$ where $V_O \subseteq V$ and $E_O \subseteq E$. We evaluate six popular GNNs (GCN, GraphSage, GAT, MoNet, GatedGCN and GIN) with four traversal-based graph sampling methods (BFS, FFS, RW, and MHRW). The six GNN models are selected according to performance and popularity; moreover, they cover all three categories of GNN models: isotropic (GCN, GraphSage), anisotropic (GAT, MoNet, GatedGCN) and Weisfeiler-Lehman (GIN) GNNs (*Dwivedi et al., 2020*). We test only traversal-based sampling methods for two reasons: these methods are practical in real settings (*Hu & Lau, 2013*), and these methods extract connected subgraphs, which is a prerequisite for GNNs. In graph sampling, we iteratively pick nodes and edges starting from a random seed node until the cardinality of the sampled node set $V_O$ reaches a given number. Apart from the original sampled subgraph $G_O(V_O, E_O, X_O)$, we also induce $V_O$ to form $G'_O(V_O, E'_O, X_O)$, i.e., $E'_O = (u, v)|u, v \in V_O, (u, v) \in E. G'_O$ has the same edges as $G$ between the vertices in $V_O$; hence, $G'_O$ can be considered as a completion of $G_O$.

We follow the principles of *Dwivedi et al. (2020)* and develop a standardized training, validation, and testing procedure for all models for fair comparisons.

In addition, we considered multilayer perceptron (MLP) as a baseline model, which utilizes only node attributes without graph structures.

## EXPERIMENTS

### Datasets

In our benchmark, we used nine datasets including six social networks (Cora, CiteSeer, PubMed (*Yang, Cohen & Salakhutdinov, 2016*), Actor (*Pei et al., 2020*), ARXIV and COLLAB (*Hu et al., 2020*)), two super-pixel networks of images (MNIST, CIFAR10 (*Dwivedi et al., 2020*)) and one artificial network generated from Stochastic Block Model (CLUSTER (*Dwivedi et al., 2020*)). Statistics for all datasets are shown in Table 1. We treated all the networks as undirected and only considered the largest connected component, moreover, we ignored edge features in our experiments.

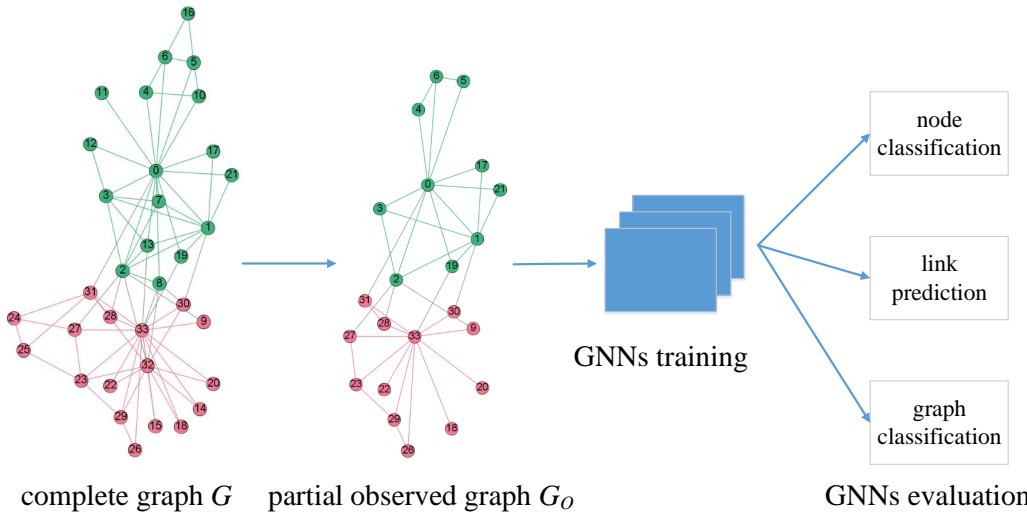

**complete graph** $G$  **partial observed graph** $G_O$  **GNNs evaluation**

**Figure 1**  **Description of GNNs evaluation under graph sampling scenarios.**

**Table 1**  **Dataset statistics.** Only the largest connected component is considered. NC, LP and GC are short for node classification, link prediction and graph classification, respectively.

| Dataset | #Graphs | Total #Nodes | Total #Edges | #Node Features | Task | Metric |
|---|---|---|---|---|---|---|
| Actor | 1 | 6,208 | 14,891 | 932 | | Weighted F1 for NC; ROC |
| CiteSeer | 1 | 2,120 | 3,679 | 3,073 | NC, LP | |
| Cora | 1 | 2,485 | 5,069 | 1,433 | | |
| PubMed | 1 | 19,717 | 44,324 | 500 | | |
| ARXIV | 1 | 139,065 | 1,085,657 | 128 | NC | Accuracy |
| COLLAB | 1 | 232,865 | 961,883 | 128 | LP | Hit@50 |
| MNIST | 70,000 | 4,939,668 | 23,954,305 | 1 | GC | Accuracy |
| CIFAR10 | 60,000 | 7,058,005 | 33,891,607 | 3 | | |
| CLUSTER | 12,000 | 1,406,436 | 25,810,340 | 7 | NC | Weighted F1 |

## Setup

Setups for our experiments are summarized in Table 2. All datasets were split into training, validation, and testing data. For node classification tasks, Cora, CiteSeer and PubMed were split according to *Yang, Cohen & Salakhutdinov (2016)*, first of the 10 splits from *Pei et al. (2020)* was picked for Actor, and CLUSTER was split according to *Dwivedi et al. (2020)*; For link prediction tasks, we used a random 70%/10%/20% training/validation/test split for positive edges in all datasets; For graph classification tasks, the splits were derived from *Dwivedi et al. (2020)*.

In GNNs, all models had a linear transform for node attributes $X$ before hidden layers. The number of hidden layers $L$ was set to $L = 2$ to avoid over-smoothing for small-scale datasets such as Actor, Core, CiteSeer and Pubmed, and we set to $L = 3$ for ARXIV and COLLAB, $L = 4$ to MNIST, CIFAR10 and CLUSTER. We added residual connections between GNN layers for medium-scale datasets (*i.e.*, ARXIV, COLLAB, MNIST, CIFAR10

**Table 2 Experiment setups.** NC, LP, LR are short for node classification, link prediction and learning rate, respectively.

| Dataset | Training/validation/ test ratio (%) | Model | | Optimizer | | LR reduce patience |
|---|---|---|---|---|---|---|
| | | #Layers | Residual | LR | Weight decay | |
| Actor | 48.0/32.0/20.0 (NC) | | | | | |
| | 70.0/10.0/20.0 (LP) | | | | | |
| CiteSeer | 3.8/15.5/31.3 (NC) | | | | | |
| | 70.0/10.0/20.0 (LP) | 2 | No | 0.01 | 5e−4 | 50 |
| Cora | 4.9/18.5/36.8 (NC) | | | | | |
| | 70.0/10.0/20.0 (LP) | | | | | |
| PubMed | 0.3/2.5/5.1 (NC) | | | | | |
| | 70.0/10.0/20.0 (LP) | | | | | |
| ARXIV | 51.9/18.3/29.8 | 3 | | 0.01 | 5e−4 | 50 |
| COLLAB | 70.0/10.0/20.0 | 3 | | | | 10 |
| MNIST | 78.6/7.1/14.3 | 4 | Yes | | | 10 |
| CIFAR10 | 75.0/8.3/16.7 | 4 | | 0.001 | 0 | 10 |
| CLUSTER | 83.3/8.3/8.3 | 4 | | | | 5 |

and CLUSTER) as suggested by *Dwivedi et al. (2020)*. We chose the hidden dimension and the output dimension that made the number of parameters almost equal for each model. The number of attention heads of GAT was set to 8, and the mean aggregation function in GraphSage was adopted. In MoNet, we set the number of Gaussian kernels to 3, and used the degrees of the adjacency nodes as the input pseudo-coordinates, as proposed in *Monti et al. (2017)*.

We used the same training procedure for all GNN models for a fair comparison. Specifically, the maximum number of training epochs was set to 1,000, and we adopted Glorot (*Glorot & Yoshua, 2010*) and zero initialization for the weights and biases, respectively. Also, we applied the Adam (*Kingma & Ba, 2015*) optimizer, and we reduced learning rate with a factor of 0.5 when a validation metric has stopped improving after the given reduce patience. Furthermore, we stopped the training procedure early if (a) learning rate was less than 1e-5, or (b) validation metric did not increase for 100 consecutive epochs, or (c) training time was more than 12 h. All model parameters were optimized with cross-entropy loss when $G_O$ was sampled.

We implemented all the six models by the Pytorch Geometrics library (*Fey & Lenssen, 2019*) and the four graph sampling methods based on *Rozemberczki, Kiss & Sarkar (2020)*.

## RESULTS

For each dataset, sampling method, and GNN model, we performed 4 runs with 4 different seeds, then reported the average metric. To answer Q1, we show means, $\mu$, and standard deviations, $\delta$, of metrics for all datasets with sampling ratio $r = |V_s|/|V| \in [0.1, 0.5]$ using GCN and MHRW (Table 3). It is worth to mention that the other GNN models and graph sampling methods had similar results. There are a few observations to be made. First, the means, $\mu$, increase and the standard deviations, $\delta$, decrease as the sampling ratio increases

in node classification and graph classification tasks, which aligns with our intuition. Second, the performance is acceptable in most single graph datasets when $r$ is relatively large, *e.g.*, compared to the complete cases, the relative losses $\Delta = 1 - \mu_r/\mu_{\text{complete}}$ are all less than 15% for CiteSeer, Cora, Pubmed, ARXIV and COLLAB when $r \geq 0.4$. This is partly because the nodes in $G_O$ have acquired sufficient neighborhood structure to accomplish the messaging and aggregation needed by GNNs. Therefore, we can still use GNNs in most single graph datasets under sampling scenarios, as long as the sampling ratio, $r$, is chosen properly. The choice of the appropriate $r$ varies depending on the dataset, sampling method, and GNN model. For example, in order to make $\Delta \leq 10\%$ on node classification tasks, the sampling ratio should satisfy $r \geq 0.5$ for Actor and PubMed, $r \geq 0.4$ for Cora, and $r \geq 0.1$ for CiteSeer. By contrast, the performance degradation is severe for multi-graph datasets (*i.e.*, CLUSTER, MNIST, CIFAR10), which is mainly due to the fact that independent random sampling destroys the intrinsic association between graphs. Hence, we cannot directly use GNNs with independent random sampling scenarios.

To answer Q2, we show $\mu$ and $\delta$ for all datasets when we fix $r = 0.3$ in Table 4.

According to Table 4, the best performing GNN model(s) is consistent across different sampling methods for a specific dataset, especially in node classification tasks, *e.g.*, GatedGCN for Actor, GCN for Cora, CiteSeer, and PubMed. The consistency suggests that datasets have a strong preference for a specific GNN model, and there is no silver-bullet GNN for all datasets. Another observation is that, some datasets show a tendency towards sampling methods, *e.g.*, BFS for Actor, RW for ARXIV. To compare all GNN models and sampling methods, we consider the relative metric score, as proposed in *Shchur et al. (2018)*. That is, for GNN models, we take the best $\mu$ from four sampling methods as 100% for each dataset, and the score of each model is divided by this value, then the results for each model are averaged over all datasets and sampling methods. We also rank GNN models by their performance (1 for best performance, 7 for worst), and compute the average rank for each model. Similarly, we calculate the score of each sampling method. The final scores for GNN models and sampling methods are summarized in Table 5. These results provide a reference for the selection of sampling methods, and a guidance for sampling-based GNN training like GraphSAINT (*Zeng et al., 2019*).

GNNs outperform MLP on average in Table 5, and this confirms the superiority of GNNs, which combine structural and attribute information, compared to methods that consider only attributes. On small datasets, GCN is the best GNN model , which proves that simple methods often outperforms more sophisticated ones (*Dwivedi et al., 2020*; *Shchur et al., 2018*). In addition, BFS is found to be the best sampling method for small datasets, partly because it samples node labels more uniformly than other methods. Figure 2 shows a comparison of the Kullback–Leibler divergence between label distributions of training and testing from different sampling methods in PubMed (NC); it can be seen that BFS has a lower score, which leads to better generalization power in GNNs. On medium datasets, the best GNN model changes to GAT, and the most competitive sampling method are RW and MHRW. This may be due to the fact that RW and MHRW can obtain a more macroscopic structure compared to BFS and FFS.

**Table 3   The means and standard deviations of metrics ($\mu \pm \delta$ (%)) for all nine datasets with sampling ratio $r \in [0.1, 0.5]$ using GCN and MHRW.** Metric for complete network structures are reported as "complete".

| | | | Node classification (GCN, MHRW) | | | |
|---|---|---|---|---|---|---|
| $r$ | Actor (NC) | CiteSeer (NC) | Cora (NC) | Pubmed (NC) | ARXIV | CLUSTER |
| 0.10 | 25.2 ± 3.6 | 63.5 ± 14.6 | 63.5 ± 11.6 | 56.6 ± 11.2 | 59.0 ± 1.2 | 31.9 ± 0.6 |
| 0.20 | 26.4 ± 2.7 | 65.8 ± 8.6 | 64.8 ± 6.0 | 62.6 ± 7.1 | 61.6 ± 1.4 | 27.3 ± 0.2 |
| 0.30 | 26.8 ± 2.4 | 64.8 ± 7.6 | 68.4 ± 4.4 | 66.6 ± 7.1 | 62.2 ± 1.2 | 26.9 ± 0.2 |
| 0.40 | 26.9 ± 1.8 | 66.6 ± 5.0 | 70.8 ± 3.7 | 70.6 ± 3.9 | 63.3 ± 1.2 | 26.5 ± 0.4 |
| 0.50 | 27.5 ± 1.6 | 66.9 ± 4.3 | 72.8 ± 3.1 | 72.4 ± 2.6 | 63.3 ± 1.5 | 26.6 ± 0.2 |
| complete | 30.1 ± 0.7 | 70.6 ± 1.0 | 78.6 ± 1.1 | 78.7 ± 0.2 | 71.4 ± 0.8 | 55.7 ± 1.6 |

| | | Link prediction (GCN, MHRW) | | | |
|---|---|---|---|---|---|
| $r$ | Actor (LP) | CiteSeer (LP) | Cora (LP) | Pubmed (LP) | COLLAB |
| 0.10 | 72.1 ± 3.9 | 91.8 ± 1.3 | 84.6 ± 5.3 | 75.6 ± 1.0 | 85.1 ± 2.3 |
| 0.20 | 69.6 ± 4.4 | 93.6 ± 2.4 | 88.3 ± 2.1 | 79.4 ± 2.0 | 81.8 ± 6.3 |
| 0.30 | 74.4 ± 2.0 | 92.9 ± 4.9 | 95.1 ± 1.7 | 88.4 ± 9.7 | 80.8 ± 1.9 |
| 0.40 | 74.2 ± 9.2 | 96.3 ± 0.6 | 90.9 ± 4.4 | 89.4 ± 9.5 | 72.5 ± 5.4 |
| 0.50 | 75.3 ± 6.8 | 94.7 ± 3.1 | 94.2 ± 3.6 | 98.5 ± 0.4 | 75.0 ± 5.5 |
| complete | 94.5 ± 0.7 | 98.7 ± 0.4 | 98.3 ± 0.6 | 99.4 ± 0.1 | 62.3 ± 5.2 |

| | Graph classification (GCN, MHRW) | |
|---|---|---|
| $r$ | MNIST | CIFAR10 |
| 0.10 | 18.6 ± 0.1 | 25.1 ± 0.8 |
| 0.20 | 18.2 ± 0.6 | 28.6 ± 0.4 |
| 0.30 | 20.0 ± 0.5 | 30.6 ± 0.5 |
| 0.40 | 22.4 ± 0.8 | 32.8 ± 0.6 |
| 0.50 | 25.3 ± 0.2 | 34.7 ± 0.4 |
| complete | 83.8 ± 0.5 | 41.9 ± 0.8 |

To answer Q3, we considered the induced subgraph $G'_O$ as a completion of $G_O$. We chose the preferred GNN model for each dataset, *e.g.*, GatedGCN for Actor, then computed the induced relative metric improvement percent as $\tau = \mu'_r / \mu_r - 1$. Figure 2 shows the improvements on all datasets with $r \in 0.1, 0.3, 0.5$.

From Fig. 3 it can be seen that network completion can improve performance in most cases. Comparing Figs. 3A, 3B and 3C shows that the induced improvement $\tau$ increases as the sampling ratio $r$ decreases especially when we perform MHRW or RW, which indicates the necessity of network completion when $\tau$ is low.

On the other hand, Fig. 3 reveals the complexity of datasets under sampling scenarios, which indicates that network completion is not always effective. Some datasets benefit from network completion in all cases, *e.g.*, Cora (NC), ARXIV and MNIST; and there are also some datasets seem to be unaffected by completion, *e.g.*, PubMed (LP) when $r \in 0.3, 0.5$ (see Figs. 3B–3C); what is more, network completion has side effects on datasets such as COLLAB. The complexity may be partly explained by structure noise in network. It

**Table 4 The means and standard deviations of metrics (μ± δ (%)) for all nine datasets with sampling ratio $r = 0.3$.** NC, LP are short for node classification and link prediction, respectively. For each dataset and graph sampling method, the best metric is marked in bold. For each dataset and GNN method, the best metric is shown in red.

| Dataset | Sampling method | GAT | GCN | GIN | Gated | Sage | MoNet | MLP |
|---|---|---|---|---|---|---|---|---|
| Actor (NC) | BFS | 29.2 ± 2.5 | 31.1 ± 2.8 | 30.1 ± 2.2 | **37.6 ± 5.8** | 34.0 ± 3.7 | 31.7 ± 5.3 | 26.6 ± 7.6 |
| | FFS | 24.9 ± 1.9 | 25.8 ± 2.1 | 25.9 ± 2.0 | **41.2 ± 2.5** | 32.3 ± 4.0 | 31.7 ± 5.0 | 29.2 ± 3.8 |
| | MHRW | 25.2 ± 2.5 | 26.8 ± 2.4 | 26.4 ± 2.4 | **40.6 ± 2.1** | 32.5 ± 2.9 | 31.5 ± 4.9 | 29.8 ± 3.1 |
| | RW | 25.5 ± 2.7 | 26.8 ± 2.6 | 27.3 ± 2.7 | **38.1 ± 4.5** | 32.9 ± 4.3 | 31.5 ± 5.1 | 30.4 ± 5.1 |
| CiteSeer (NC) | BFS | 71.0 ± 7.7 | **73.5 ± 4.3** | 67.5 ± 6.3 | 52.5 ± 27.6 | 65.4 ± 17.6 | 60.5 ± 21.5 | 63.1 ± 12.7 |
| | FFS | 65.2 ± 6.7 | **67.4 ± 5.1** | 63.9 ± 6.6 | 53.8 ± 24.6 | 61.4 ± 14.6 | 54.5 ± 17.5 | 57.4 ± 11.8 |
| | MHRW | 61.3 ± 8.8 | **64.8 ± 7.6** | 59.6 ± 9.8 | 49.1 ± 23.5 | 54.8 ± 11.8 | 43.6 ± 13.1 | 52.3 ± 12.6 |
| | RW | 64.3 ± 7.0 | **66.8 ± 6.2** | 61.4 ± 7.9 | 56.3 ± 9.7 | 58.0 ± 14.1 | 48.9 ± 13.8 | 50.7 ± 12.2 |
| Cora (NC) | BFS | 64.7 ± 8.0 | **68.6 ± 7.0** | 61.7 ± 7.6 | 48.0 ± 14.0 | 60.6 ± 9.7 | 50.6 ± 11.8 | 59.4 ± 9.5 |
| | FFS | 61.8 ± 6.5 | **66.6 ± 4.7** | 62.3 ± 5.6 | 54.5 ± 10.5 | 55.5 ± 10.5 | 45.4 ± 8.5 | 56.2 ± 10.9 |
| | MHRW | 61.9 ± 5.4 | **68.4 ± 4.4** | 64.1 ± 4.8 | 53.5 ± 8.8 | 54.5 ± 10.6 | 42.7 ± 8.1 | 56.5 ± 8.9 |
| | RW | 62.1 ± 5.7 | **69.4 ± 5.0** | 62.8 ± 6.5 | 49.7 ± 12.0 | 53.8 ± 10.5 | 42.7 ± 8.9 | 54.2 ± 8.9 |
| PubMed (NC) | BFS | 71.9 ± 9.1 | **74.4 ± 5.5** | 69.8 ± 7.9 | 55.5 ± 19.0 | 65.9 ± 19.2 | 63.6 ± 19.8 | 63.7 ± 19.9 |
| | FFS | 65.4 ± 8.4 | **68.5 ± 5.8** | 66.9 ± 6.2 | 66.3 ± 4.5 | 61.6 ± 15.0 | 52.0 ± 18.4 | 60.0 ± 13.1 |
| | MHRW | 63.0 ± 8.7 | **66.6 ± 7.1** | 64.6 ± 6.1 | 52.1 ± 16.8 | 52.7 ± 14.8 | 50.9 ± 12.7 | 54.4 ± 13.0 |
| | RW | 64.4 ± 7.8 | **67.7 ± 4.9** | 65.8 ± 5.5 | 53.7 ± 21.3 | 57.8 ± 14.4 | 53.5 ± 14.1 | 56.1 ± 13.8 |
| ARXIV | BFS | 57.4 ± 2.2 | 55.7 ± 2.4 | **61.6 ± 3.6** | 60.8 ± 2.1 | 59.6 ± 2.1 | 56.9 ± 2.0 | 58.9 ± 1.8 |
| | FFS | 61.5 ± 4.1 | 60.5 ± 0.6 | **62.9 ± 2.8** | 59.5 ± 4.3 | 59.2 ± 3.5 | 58.2 ± 4.2 | 53.3 ± 4.0 |
| | MHRW | 61.9 ± 2.6 | 62.2 ± 1.2 | **64.1 ± 1.2** | 59.2 ± 1.0 | 60.2 ± 0.9 | 58.6 ± 1.1 | 52.5 ± 1.2 |
| | RW | 64.6 ± 1.6 | 65.7 ± 0.5 | 67.3 ± 0.6 | 64.2 ± 0.8 | 63.1 ± 0.3 | 63.4 ± 1.6 | 57.7 ± 0.3 |
| CLUSTER | BFS | 26.4 ± 0.5 | 26.3 ± 0.6 | 26.1 ± 1.2 | 25.6 ± 1.2 | 24.5 ± 0.3 | 26.3 ± 1.2 | **29.3 ± 0.5** |
| | FFS | 27.2 ± 0.7 | 26.1 ± 0.5 | 23.8 ± 0.4 | 25.1 ± 1.3 | 25.4 ± 0.3 | 25.0 ± 1.0 | **29.1 ± 0.4** |
| | MHRW | 27.6 ± 0.4 | 26.9 ± 0.2 | 25.6 ± 1.5 | 25.1 ± 0.7 | 25.9 ± 0.5 | 26.6 ± 0.6 | **29.5 ± 0.2** |
| | RW | 27.3 ± 0.4 | 26.8 ± 0.4 | 24.4 ± 1.1 | 25.3 ± 1.0 | 26.7 ± 0.2 | 26.2 ± 1.0 | **28.9 ± 0.4** |
| Actor (LP) | BFS | 91.0 ± 17.3 | 99.1 ± 0.6 | 97.5 ± 2.9 | 99.3 ± 0.7 | **99.9 ± 0.0** | 99.8 ± 0.1 | 91.1 ± 1.5 |
| | FFS | 61.2 ± 15.2 | 71.6 ± 1.7 | 79.3 ± 3.3 | **86.6 ± 4.8** | 63.1 ± 18.2 | 81.8 ± 5.8 | 51.8 ± 1.8 |
| | MHRW | 63.7 ± 14.1 | 74.4 ± 2.0 | 75.9 ± 2.0 | **84.9 ± 4.8** | 49.3 ± 3.4 | 81.6 ± 4.6 | 50.6 ± 1.6 |
| | RW | 56.2 ± 11.5 | 73.8 ± 7.5 | 81.4 ± 6.0 | **84.4 ± 4.2** | 81.2 ± 4.2 | 87.3 ± 3.1 | 57.0 ± 1.0 |
| CiteSeer (LP) | BFS | 91.8 ± 3.8 | 91.9 ± 4.8 | 95.1 ± 2.4 | 95.2 ± 1.9 | **95.6 ± 3.0** | 92.2 ± 10.8 | 72.0 ± 10.2 |
| | FFS | 83.5 ± 20.6 | 91.6 ± 5.1 | 90.4 ± 2.7 | 94.3 ± 0.5 | 94.7 ± 2.3 | **95.7 ± 2.5** | 59.9 ± 4.6 |
| | MHRW | 90.6 ± 1.8 | 92.9 ± 4.9 | 92.4 ± 1.0 | 94.0 ± 1.6 | **94.4 ± 2.2** | 91.6 ± 6.5 | 57.4 ± 4.0 |
| | RW | 85.4 ± 18.7 | 93.1 ± 4.7 | 92.8 ± 2.1 | 92.9 ± 2.0 | 93.6 ± 5.9 | **96.4 ± 0.9** | 61.8 ± 3.9 |
| Cora (LP) | BFS | 91.5 ± 4.1 | 94.5 ± 3.3 | 94.3 ± 2.3 | 97.9 ± 2.2 | **98.0 ± 1.2** | 96.7 ± 0.8 | 83.6 ± 2.3 |
| | FFS | **95.4 ± 0.8** | 92.1 ± 4.6 | 91.3 ± 1.7 | 90.1 ± 3.3 | 90.1 ± 5.4 | 93.6 ± 3.3 | 54.8 ± 0.6 |
| | MHRW | 80.0 ± 19.3 | **95.1 ± 1.7** | 92.3 ± 1.0 | 90.1 ± 3.7 | 93.8 ± 3.4 | 93.3 ± 3.8 | 57.3 ± 7.1 |
| | RW | 89.7 ± 4.8 | 92.0 ± 3.5 | 91.6 ± 5.0 | 91.9 ± 3.4 | **92.7 ± 3.1** | 86.8 ± 10.4 | 65.7 ± 6.6 |
| PubMed (LP) | BFS | 84.8 ± 20.1 | 98.2 ± 1.7 | 99.1 ± 0.6 | **99.7 ± 0.2** | 99.1 ± 0.3 | 99.2 ± 0.8 | 78.9 ± 2.8 |
| | FFS | 83.0 ± 18.9 | 90.1 ± 8.1 | 98.1 ± 0.8 | 98.2 ± 0.7 | 63.7 ± 19.7 | **98.7 ± 0.6** | 81.3 ± 1.9 |
| | MHRW | 84.6 ± 20.0 | 88.4 ± 9.7 | 98.6 ± 0.4 | **98.6 ± 0.8** | 74.2 ± 23.9 | 97.9 ± 2.1 | 74.0 ± 14.0 |
| | RW | 96.2 ± 0.8 | 89.0 ± 9.1 | 98.3 ± 0.6 | 98.7 ± 0.8 | 97.1 ± 0.0 | **99.0 ± 1.0** | 84.7 ± 1.0 |

**Table 4** (*continued*)

| Dataset | Sampling method | GAT | GCN | GIN | Gated | Sage | MoNet | MLP |
|---|---|---|---|---|---|---|---|---|
| COLLAB | BFS | **98.9** ± 0.2 | 9.3 ± 1.6 | 11.3 ± 1.1 | 11.1 ± 1.1 | 78.3 ± 43.0 | 9.6 ± 1.1 | 4.5 ± 2.6 |
| | FFS | 64.7 ± 31.0 | 62.2 ± 5.1 | 29.5 ± 5.6 | 30.6 ± 19.2 | **88.6** ± 8.5 | 55.2 ± 7.9 | 7.7 ± 1.9 |
| | MHRW | 70.5 ± 34.8 | 80.8 ± 1.9 | 44.5 ± 13.3 | 37.7 ± 12.5 | 89.6 ± 5.7 | 56.4 ± 11.6 | 18.4 ± 2.5 |
| | RW | **77.5** ± 9.2 | 72.0 ± 1.8 | 42.3 ± 5.0 | 33.6 ± 6.7 | 74.3 ± 11.4 | 60.9 ± 6.1 | 10.7 ± 4.3 |
| MNIST | BFS | 24.8 ± 0.1 | 25.3 ± 0.4 | 24.9 ± 0.8 | 26.9 ± 0.4 | 24.6 ± 0.9 | **27.3** ± 0.6 | 21.3 ± 0.6 |
| | FFS | 23.3 ± 0.9 | 21.2 ± 0.2 | 21.8 ± 1.5 | **24.6** ± 1.2 | 22.4 ± 1.3 | 23.1 ± 0.2 | 20.9 ± 0.4 |
| | MHRW | 21.4 ± 0.2 | 20.0 ± 0.5 | 21.9 ± 0.4 | 21.9 ± 0.2 | 21.5 ± 0.8 | **22.4** ± 0.3 | 20.1 ± 1.1 |
| | RW | 21.9 ± 0.1 | 21.0 ± 0.5 | 21.1 ± 2.3 | **22.3** ± 0.3 | 22.1 ± 0.5 | 22.1 ± 0.2 | 20.7 ± 0.3 |
| CIFAR10 | BFS | 33.0 ± 0.4 | 30.6 ± 0.5 | 28.3 ± 1.1 | 33.3 ± 1.0 | 33.0 ± 0.4 | **34.0** ± 0.7 | 32.8 ± 0.5 |
| | FFS | **34.6** ± 0.4 | 29.7 ± 0.1 | 20.2 ± 2.1 | 34.5 ± 0.6 | 33.7 ± 1.1 | 34.3 ± 0.6 | 32.8 ± 0.5 |
| | MHRW | **36.5** ± 0.1 | 30.6 ± 0.5 | 21.5 ± 2.1 | 35.5 ± 0.2 | 35.6 ± 0.7 | 27.9 ± 1.5 | 33.5 ± 0.5 |
| | RW | 35.4 ± 0.3 | 31.2 ± 0.3 | 29.4 ± 3.2 | 35.4 ± 0.4 | 34.8 ± 0.2 | **35.7** ± 0.7 | 33.7 ± 0.3 |

is evident that removing task-irrelevant edges from original structure can improve GNN performance (*Luo et al., 2021*; *Zheng et al., 2020*). We treat graph sampling as a structural denoising process. If the original network $G$ has only a small amount of structure noise, completion restores the informative edges removed by sampling, thus improving the GNN performance. Whereas if the structure noise is large in $G$, completion weakens the denoising effect of sampling and leads to performance degradation.

## CONCLUSIONS

We focused on the performance of GNNs with partial observed network structure. By treating the incomplete structure as one of the many graphs generated by a certain sampling process, we determined the robustness of GNNs in a statistical way via multiple independent random sampling. Specifically, we performed an empirical evaluation of six state-of-the-art GNNs on three network learning tasks (*i.e.*, node classification, link prediction and graph classification) with four popular graph sampling methods. We confirmed that GNNs can still be applied under graph sampling scenarios in most single graph datasets, but not on multiple graph datasets. We also identified the best GNN model and sampling method, that is, GCN and BFS for small datasets, GAT and RW for medium datasets. Which provides a guideline for future applications. Moreover, we found that network completion can improve GNN performance in most cases, however, specific analysis is needed case by case due to the complexity of datasets under sampling scenarios. Thus, suggesting that completion and denoising should be done with careful evaluation. We hope this work, along with the public codes, will encourage future works on understanding the relationship between structural information and GNNs.

**Table 5   Relative metric score and average rank for (a) GNNs on small datasets, (b) graph sampling methods on small datasets, (c) GNNs on medium datasets, and (d) graph sampling methods on medium datasets.**

| (a) | | |
|---|---|---|
| **GNN** | **Relative metric (%)** | **Rank** |
| GCN | 93.3 | 2.9 |
| GIN | 92.2 | 3.4 |
| GatedGCN | 91.2 | 3.6 |
| MoNet | 89.5 | 3.6 |
| GraphSage | 87.6 | 3.7 |
| GAT | 87.4 | 4.6 |
| MLP | 73.1 | 6.3 |

| (b) | | |
|---|---|---|
| **Sampling method** | **Relative metric (%)** | **Rank** |
| BFS | 98.4 | 1.4 |
| RW | 90.7 | 2.6 |
| FFS | 90.3 | 2.7 |
| MHRW | 87.9 | 3.2 |

| (c) | | |
|---|---|---|
| **GNN** | **Relative metric (%)** | **Rank** |
| GAT | 94.1 | 2.6 |
| GraphSage | 93.5 | 3.8 |
| GCN | 85.9 | 4.3 |
| MoNet | 85.5 | 3.8 |
| GatedGCN | 82.4 | 3.9 |
| GIN | 77.3 | 4.6 |
| MLP | 75.9 | 5.2 |

| (d) | | |
|---|---|---|
| **Sampling method** | **Relative metric (%)** | **Rank** |
| RW | 93.7 | 2.1 |
| MHRW | 93.1 | 2.3 |
| FFS | 89.1 | 3.0 |
| BFS | 85.3 | 2.7 |

## Funding
The authors received no funding for this work.

## Competing Interests
The authors declare there are no competing interests.

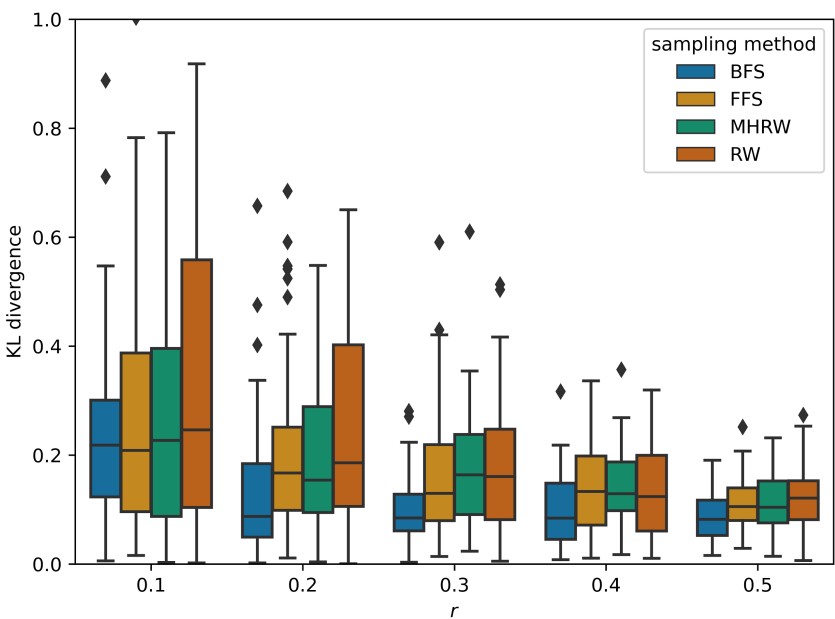

**Figure 2** Kullback–Leibler divergence between label distributions of training and testing on Pubmed (NC).

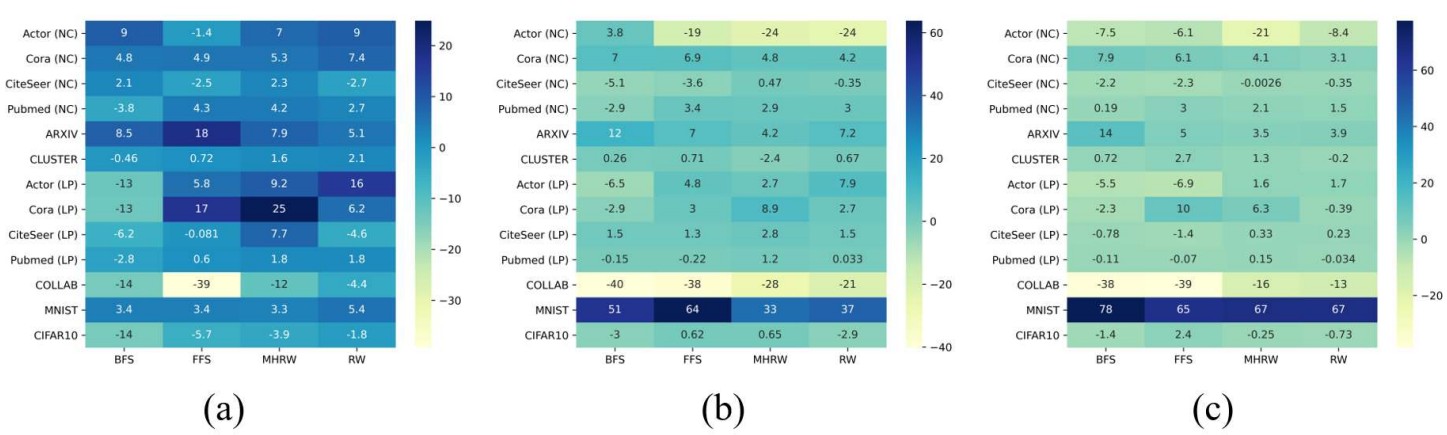

**Figure 3** Induced relative metric improvement for (A) $r = 0.1$, (B) $r = 0.3$, and (C) $r = 0.5$.

## Author Contributions

- Qiang Wei conceived and designed the experiments, performed the experiments, analyzed the data, performed the computation work, prepared figures and/or tables, authored or reviewed drafts of the paper, and approved the final draft.
- Guangmin Hu conceived and designed the experiments, authored or reviewed drafts of the paper, and approved the final draft.

## Data Availability

The data is available at the torch geometric library and Open Graph Benchmark (OGB).

- Actor is available at GitHub: https://github.com/graphdml-uiuc-jlu/geom-gcn/tree/master/new_data/film

- CiteSeer, Cora and Pubmed is available at GitHub: https://github.com/kimiyoung/planetoid/raw/master/data

- ARXIV: http://snap.stanford.edu/ogb/data/nodeproppred

- COLLAB: http://snap.stanford.edu/ogb/data/linkproppred

- CLUSTER, MNIST, CIFAR10: Wei Qaing. (2022). Three Datasets Cloned from GNNBenchmarkingDatasets (Version v2) [Data set]. Zenodo. https://doi.org/10.5281/zenodo.6050722

Our code is available at GitHub: https://github.com/weiqianglg/evaluate-GNNs-under-graph-sampling.

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
