# Peer review of "Evaluating graph neural networks under graph sampling scenarios"

_PeerJ Computer Science, doi:10.7717/peerj-cs.901_

## Round 0.1 · original submission · Major Revisions

Two reports have been received for this paper. I recommend a revision before further processing. Please note that I do not expect you to cite any recommended references unless they are crucial. I look forward to receiving your revised paper and a response letter.

Reviewer 1 ·

Basic reporting

This is an experimental analysis paper for evaluating different baseline GNNs using graph sampling. The motivation of the work seems to inform the community of a experimental benchmark that supplies the trends of using different sampling methods while using several GNN instances. The writing of the paper is clear and the experiments are fairly compared when the models are evaluated on the 4 datasets considered. The code framework is public and from my observation it follows a similar layout/setting as Benchmarking GNNs [Dwivedi et al., 2020] following the priciples of fair GNN benchmarking. However, the paper has no technical contribution apart from the aforementioned sentences; and this is obvious from the introduction and setting of the paper.

Following are my concerns which may prohibit the usefulness of this paper as a benchmark of sampling methods using different GNNs:

* The datasets used are quite small and there is a well-known agreement in the community to move on to more large and complex datasets than Cora, Citeseer, etc. that are used in the paper. See Dwivedi et al., 2020, Open Graph Benchmark: Hu et al., 2020 which describe the need to using other datasets than aforementioned to evaluate GNN trends fairly. It is also clear that this paper is aware of the previous works on GNN benchmarks. Hence, I do not understand the need of making a graph sampling benchmark using the same small datasets!
* The paper follows the setting of Dwivedi et al., 2020 but do not consider GatedGCNs which were best performing in that work. This may be question the validity of the paper's insight where it is written that GCN/BFS are generally the best performing (L65), since a previous well performing GNN baseline-GatedGCN is not considered for evaluation in this work.
* If the paper's intent was to establish a evaluation trend of different GNN under sampling, it is necessary to consider robustness of experiments to make the results reliable and useful for the community. While the experiments in the paper are fair and unquestionable, it is not necessarily robust. For example, it does not seem the results are reliable if they are not evaluated on more complex and diverse datasets, and also on diverse tasks (such as graph prediction, link prediction, node prediction). This work only considers node prediction on small datasets.

Experimental design

* The research questions are well defined and meaningful. If the work would be more robust, it could fill the research gaps. However, see my concerns above in "Basic Reporting" which may not make this paper robust.
* Of the experiments considered in the paper, the settings are well defined for reproducibility. Similarly, the methods are sufficicently detailed to replicate. However, the concerns are with the datasets used and the lack of robustness in such an experimental work. Please refer to comments in "Basic Reporting" section on this.

Validity of the findings

* Of the experiments performed in the paper, the underlying data is provided, but are questionable given the current state and trend of the graph deep learning field in general. In particular, the community is slowly transitioning to evaluate GNNs on more diverse, complex and medium-to-large scale datasets. The datasets used in the paper are small and I believe that the paper is aware of that.

Additional comments

* In the writing of the paper, several references do not mention date and have "n.d." instead of the date. For eg. Bruna et al. n.d. It seems this is a minor thing which the paper should have taken care of before submission.

·

Basic reporting

The authors provide a systematic review of graph based methods. They tried to identify the performance of graph neural network models for graphlets. Authors performed the evaluation of five Graph Neural Networks. The idea is good and interesting. However I have few comments on this article.

Experimental design

1- In experimental setup, author did not mention how many hidden layers are in the network.
2- What information/features are you extracting in layers of the network.
3- Please mention the no of training, testing and validation samples you are using in the experiments.
4- can you elaborate row no 138 "there is no missing edges among..." why ?
5- what are effects you noticed for 1000 epochs for small and large datasets? any visualization graphs etc??
6- among five GNNs you did not considered the Siamese GNN, any reason about that? for example "Riba, Pau, et al. "Learning graph distances with message passing neural networks." 2018 24th International Conference on Pattern Recognition (ICPR). IEEE, 2018." and NI kajla et. al. "Graph Neural Networks Using Local Descriptions in Attributed Graphs: An Application to Symbol Recognition and Hand Written Character Recognition"

Validity of the findings

1- in Abstract row no 27 author say that to complete a sample subgraph does not improve the performance. they adds that it depends on the dataset, However I am not clear about the statement. Please elaborate more about it.

---

## Round 0.2 · accepted · Accept

The paper can be accepted. Congratulations.

·

Basic reporting

The authors provide a systematic review of graph-based methods. They tried to identify the performance of graph neural network models for graphlets. Authors performed the evaluation of five Graph Neural Networks. The idea is good and interesting.

Experimental design

My previous comments on the experimental portion have been addressed.

Validity of the findings

findings of this study are significant

Additional comments

All my previous comments have been addressed. I have no further comments.